

# Aerial image segmentation of embankment dams based on multispectral remote sensing: a case study in the Belo Monte Hydroelectric Complex, Pará, Brazil

Carlos André de Mattos Teixeira[1], Thabatta Moreira Alves de Araujo[2], Evelin Cardoso[1], Marcos Antonio Costantin Filho[3], João Weyl Costa[1] and Carlos Renato Lisboa Frances[1]

[1] Instituto de Tecnologia, Universidade Federal do Pará, Belém, Pará, Brazil
[2] Departamento de Computação, Centro Federal de Educação Tecnológica de Minas Gerais, Divinópolis, Minas Gerais, Brazil
[3] Norte Energia S.A., Brasília, DF, Brazil

Corresponding author
Carlos André de Mattos Teixeira, carlos.mattos@itec.ufpa.br

## ABSTRACT

Visual inspection is essential to ensure the stability of earth-rock dams. Periodic visual assessment of this type of structure through vegetation cover analysis is an effective monitoring method. Recently, multispectral remote sensing data and machine learning techniques have been applied to develop methodologies that enable automatic vegetation analysis and anomaly detection based on computer vision. As a first step toward this automation, this study introduces a methodology for land cover segmentation of earth-rock embankment dam structures within the Belo Monte Hydroelectric Complex, located in the state of Pará, northern Brazil. Random forest (RF) ensemble models were trained on manually annotated data captured by a multispectral sensor embedded in an uncrewed aerial vehicle (UAV). The main objectives of this study are to assess the classification performance of the algorithm in segmenting earth-rock dams and the contribution of non-visible band reflectance data to the overall model performance. A comprehensive feature engineering and ranking approach is presented to select the most descriptive features that represent the four dataset classes. Model performance was assessed using classical performance metrics derived from the confusion matrix, such as accuracy, Kappa coefficient, precision, recall, F1-score, and intersection over union (IoU). The final RF model achieved 90.9% mean IoU for binary segmentation and 91.1% mean IoU for multiclass segmentation. Post-processing techniques were applied to refine the predicted masks, enhancing the mean IoU to 93.2% and 91.9%, respectively. The flexible methodology presented in this work can be applied to different scenarios when treated as a framework for pixel-wise land cover classification, serving as a crucial step toward automating visual inspection processes. The implementation of automated monitoring solutions improves the visual inspection process and mitigates the catastrophic consequences resulting from dam failures.

## INTRODUCTION

The management of hydrological resources involves complex and demanding tasks, particularly when it comes to ensuring the safety of dams and dikes. These structures are crucial for supporting human activities and sustainable development. Continuous assessment of their structural condition is essential to prevent potential failures that could lead to substantial financial, social, and environmental impacts (*Solórzano et al., 2022*; *Limão, de Araújo & Frances, 2023*).

Approximately 78% of registered large dams belong to the category of earth-rock dams (*World Register of Dams, 2024*). These structures are typically constructed using a mixture of earth and rock materials (*Jung, Berges & Garrett, 2014*). The inclined surfaces of an earth-rock dam, known as slopes, are crucial for preserving its structural integrity (*Hu & Lu, 2023*). The slopes are integral components of a dam and must have their geometry preserved to maintain structural integrity (*Olson & Stark, 2003*). If these conditions are not met, the slope becomes unstable, which may lead to geodynamic events responsible for many natural disasters (*Das, 2011*; *Ledesma, Sfriso & Manzanal, 2022*; *Guan & Yang, 2020*). Identifying slope damage can provide valuable insight into the condition of dams. Some indicators may be warning signs of potential structural compromise that could lead to catastrophic failure.

Promptly detecting and interpreting early warning signals is crucial, as it provides valuable information about the condition of geotechnical structures and allows for proactive measures to prevent possible failures. This is done through continuous structural health monitoring considering constructive design, behavior models, location, and other design issues (*Li et al., 2019*; *Costigliola et al., 2022*). Dam health monitoring (DHM) can provide a more comprehensive and timely understanding of the structural health status of dams and dykes, but still presents significant challenges (*Deng et al., 2025*). Field instrumentation and visual inspection are the primary strategies for monitoring these structures.

Visual monitoring, combined with engineering expertise, is crucial for understanding the performance and conducting safety assessments of the structure (*Fanelli, 1994*). This approach is essential for detecting damage to the downstream area and surface of the slope, superficial erosion, skin holes, seepage, settlement, landslides, and abnormal vegetation growth (*Espósito & Palmier, 2013*). Experienced professionals typically analyze the environment, identify damage, and establish actions to mitigate failures. Additionally, dam inspections require long hours, exposing professionals to hazardous conditions and risks due to the remote and difficult-to-access locations of these projects (*Pan & Chen, 2015*). The employment of monitoring technologies can substantially reduce the risk to professionals and the frequency of visual inspections, while simultaneously enhancing performance and safety (*Lim et al., 2021*).

Visual inspection of the slope's vegetation coverage can reveal crucial signs of slope instability. Vegetation plays a crucial role in preventing erosion, protecting and maintaining the soil on the surface (*Cazzuffi & Crippa, 2005*). Vegetation cover reduces soil erosion rates by protecting against rain impact, reducing surface runoff, slowing down

runoff speed, and increasing soil water infiltration capacity (*Cooke & Doornkamp, 1990*). The absence of vegetation can be indicative of slope damage or anomalies, such as landslides, cracks, animal burrows, abnormal vegetation, or erosion.

Recently, advanced remote sensing (RS) technologies such as satellite imagery and UAVs have facilitated image data acquisition to help visually inspect structures (*Jang, Kim & An, 2019*). RS data collected by sensors embedded in UAVs offer advantages over satellite imagery, including higher resolutions, path planning, route optimization, height adjustment, and higher sample rates. Moreover, the presence of metadata bundled with UAV imagery, such as GPS coordinates, flight height, and position of the Sun, enables data processing to obtain valuable insights. Geolocation-based databases containing relevant information can be created through these processes.

It is well established that the human eye can only perceive the visible spectrum of light, which comprises wavelengths from 400 to 750 nanometers. Traditional digital imagery uses red (600 nm), green (546 nm), and blue (436 nm) as primary colors to form the RGB composition. However, valuable information can be extracted when using data captured from beyond the visible electromagnetic spectrum. Multispectral imagery composed of reflectance data of non-visible bands captured by satellites is a widely regarded object of research in the field of remote sensing (*Candiago et al., 2015*). In recent years, there has been growing interest in embedding such multispectral sensors in UAVs.

When it comes to monitoring earth dams, despite the lack of applications found in the related literature, the usage of UAVs equipped with multispectral sensor cameras offers a significant advantage. Their high spatial resolution, with ground sampling distances (GSD) as fine as 0.5 to 10 cm, allows the detection of subtle pigment variations and vegetation anomalies that may indicate early signs of structural instability, such as moisture accumulation, erosion, animal burrows, or stress areas that are imperceptible to the human eye (*Mamaghani & Salvaggio, 2019*). In addition, UAVs operate at lower altitudes, minimizing atmospheric interference and improving the accuracy of vegetation indices such as NDVI, NDRE, and GNDVI, which are critical to assessing plant health as a proxy for structural integrity (*Behera, Bakshi & Sa, 2023*; *Cheng et al., 2024*). Multispectral data, particularly in the near-infrared and thermal bands, have proven to be essential tools in DHM, enabling proactive maintenance and improving risk assessment strategies (*Louargant et al., 2017*; *Davidson et al., 2022*; *Lim et al., 2021*). Despite recent advancements in the field of remote sensing and the critical importance of DHM, image processing and computer vision techniques have emerged relatively late in the field, particularly for earth dams, highlighting a notable gap in the literature *Deng et al. (2025)*.

Previous studies have demonstrated that meaningful patterns can be extracted from multispectral data by combining traditional image processing techniques and machine learning algorithms (*Wu et al., 2019*). Land-cover segmentation is achieved by associating characteristic reflectance behaviors with each class (*Ahn et al., 2020*). In the context of DHM, land-cover segmentation plays a critical role in isolating vegetation and other structural elements, such as drainage channels and other structures, thereby enhancing the precision of vegetation coverage health analysis. This is particularly important as changes in vegetation patterns may indicate early-stage structural anomalies such as erosion,

seepage, or slope instability, which are key precursors to dam failure. The random forest (RF) algorithm (*Breiman, 2001*) has been widely applied in the literature for land-cover segmentation tasks (*Linhui, Weipeng & Huihui, 2021*; *Conceição et al., 2021*; *Wu et al., 2019*) and has been shown to perform well compared to algorithms such as K-nearest neighbors (KNN) and support vector machines (SVM) (*Linhui, Weipeng & Huihui, 2021*). Thus, the RF algorithm was employed in this work for pixel-wise land cover segmentation.

This study represents an initial step toward the automated monitoring of earth-rock dams using multispectral UAV imagery. It addresses a key gap in the literature by focusing on land-cover segmentation of earth dams—an area where image-based methods remain largely unexplored. Using a manually annotated dataset collected from structures within the Belo Monte Hydroelectric Complex, the study develops and evaluates Random Forest models for both binary and multi-class pixel-wise segmentation. The segmentation masks include key structural features common to 30 dams in the complex. Model performance was assessed using accuracy, Kappa coefficient, precision, recall, and F1-score. A feature-ranking method was also applied to assess the contribution of each spectral band, particularly the impact of non-visible wavelengths.

The main contributions of this study are: (1) the introduction of the first publicly available UAV-based multispectral dataset for earth dams; (2) the application of land-cover segmentation to isolate relevant vegetated areas, reducing interference from structural elements and enhancing the precision of vegetation-index-based anomaly detection; and (3) the use of a traditional machine learning approach that performs well under limited data conditions, offering a reliable and data-efficient alternative in the absence of large-scale annotated datasets.

The segmentation methodology presented in this work was developed in partnership with the enterprise as a foundational step toward the multimodal decision-making support system depicted in Fig. 1. Automated visual inspection routines facilitate precise, replicable, and on-demand data collection, enabling subsequent analysis while significantly mitigating risks and supporting decision-making processes based on different kinds of data, such as visual inspection reports, sensor data, and multispectral UAV imagery. This ultimately strengthens the safety management of these structures, helping to protect adjacent vulnerable communities and environmental resources.

A key advantage of the semantic segmentation process is its ability to isolate specific land cover classes, which is essential for anomaly detection in earth dams. In these structures, detecting subtle changes in vegetation health is critical, as they can indicate moisture accumulation, erosion, animal burrows, or early signs of structural instability. By segmenting the dam surface and removing interference from non-relevant features, subsequent analysis based on vegetation indices (*e.g.*, NDVI, NDRE, GNDVI) and anomaly detection becomes more accurate and reliable.

This article is structured as follows. "Related Work" covers the related work and main technology concepts, discussing their advantages, drawbacks, and state-of-the-art. "Materials and Methods" describes the methodology adopted in this work, including details about data collection and processing, model training, and performance assessment. The results of the land cover segmentation are shown in "Results and Discussion".

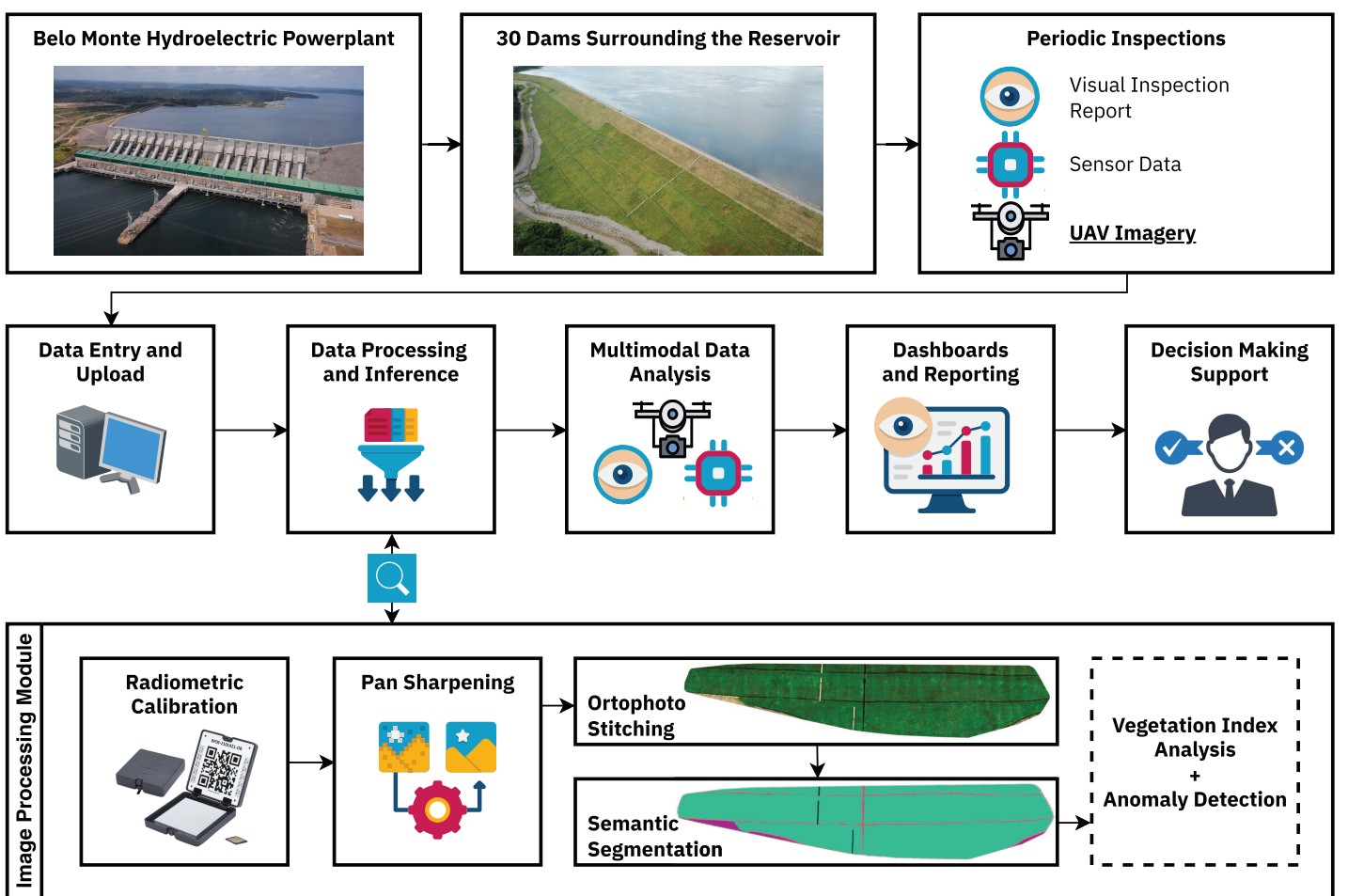

**Figure 1 Complete dam monitoring workflow.** The land-cover segmentation methodology is embedded to the Data Processing and Inference module, being responsible for differentiating between the different land-cover classes present in the earth dam structures, enabling further vegetation index and anomaly detection analyses.

"Discussion" provides a discussion of the results presented in the previous section, and "Conclusions" concludes this article by summarizing the key findings of this work and identifying areas for future research.

## RELATED WORK

This research focuses on the processing of multispectral aerial images to isolate, segment and find features that represent different areas, which can be defined as a problem of land cover segmentation (*Adam, Mutanga & Rugege, 2010*; *Wu et al., 2019*). The objective is to segment the different types of land cover found in the earth-rock dam structures located in the study area based on multispectral remote sensing data collected by UAVs. A feature extraction pipeline was developed to extract characteristics from the images, which are then fed to the machine learning model to perform a pixel-wise segmentation of the images. The resulting segmentation masks are necessary for the creation of land cover

maps, which are valuable for decision makers, as they enable further image analyses focused on specific regions.

A comprehensive analysis of land cover involves scrutinizing the spectral signatures of various objects in an image. These signatures are acquired by measuring the amount of light reflected or absorbed by each object at different wavelengths (*Louargant et al., 2017*; *Mamaghani & Salvaggio, 2019*; *Ahn et al., 2020*). Through this analysis, numerous characteristics can be identified and monitored, such as vegetation health, soil moisture, and impacts of climate change, which have significant implications in agriculture, (*Ahn et al., 2020*; *Davidson et al., 2022*; *Louargant et al., 2017*; *Tsouros, Bibi & Sarigiannidis, 2019*; *Candiago et al., 2015*), remote sensing (*Mamaghani & Salvaggio, 2019*; *Wu et al., 2019*; *Adam, Mutanga & Rugege, 2010*; *Kotaridis & Lazaridou, 2021*; *Linhui, Weipeng & Huihui, 2021*), and monitoring of environmental changes (*Zhu et al., 2020*; *Bannari & Al-Ali, 2020*; *Shobiga & Selvakumar, 2015*; *Ling et al., 2017*).

Exploring land cover solutions with the help of UAV systems for data collection in conjunction with machine learning is a promising way to ensure the safety of dams (*Wan et al., 2019*; *Davidson et al., 2022*). Dams are essential structures for human activities and, although the occurrence of failure is rare, the potential for catastrophic damage is high (*World Register of Dams, 2024*; *Guan & Yang, 2020*). Therefore, a proactive approach is necessary for risk management (*Balaniuk, Isupova & Reece, 2020*; *Wan et al., 2019*). Dams can be classified into different parts, such as downstream and upstream slopes, crest, and toe drain, depending on their purpose and construction design (*Costigliola et al., 2022*; *Das, 2011*). Exploring land cover solutions can help manage risk and ensure the safety of these structures (*Balaniuk, Isupova & Reece, 2020*).

Image processing approaches are widely employed in different fields and nowadays are usually aided by machine and deep learning techniques such as RF (*Wu et al., 2019*; *Linhui, Weipeng & Huihui, 2021*; *Conceição et al., 2021*; *Luo et al., 2022*; *Breiman, 2001*), support vector machines (SVM) (*Linhui, Weipeng & Huihui, 2021*), and convolutional neural networks (CNN) (*Balaniuk, Isupova & Reece, 2020*; *Guo et al., 2020*; *Nogueira et al., 2020*; *Bragagnolo, Da Silva & Grzybowski, 2021*; *Osco et al., 2021*; *Behera, Bakshi & Sa, 2023*; *Wang et al., 2022*). However, the processing of multispectral images in land cover applications presents a significant challenge (*Guo et al., 2020*). Some notable obstacles include the lack of available high-resolution multispectral image datasets and the preprocessing steps required to correct raw images.

The methodology presented by *Linhui, Weipeng & Huihui (2021)* for pixel-wise object-based classification of aerial and satellite images using the RF algorithm is noteworthy. The proposed approach was compared to the SVM classifier through a series of experiments. The results showed that the RF model outperformed the SVM classifier, demonstrating that the use of RF for spectral image processing can significantly enhance the accuracy of land cover classification.

In addition, the problem of land cover mapping was applied to urban environments by *Wu et al. (2019)*, presenting a valuable contribution. This study applied the RF classifier to segment multispectral images from the GF-2 satellite and LiDAR point cloud data. The

findings highlight the effectiveness of combining multispectral data from the GF-2 and LiDAR data. Another contribution of this article was the inclusion of different feature extraction techniques to compose the dataset, which includes texture characteristics (102 RF variables). A feature subset based feature selection methodology was employed in order to achieve higher classification accuracy while reducing computational complexity.

*Guo et al. (2020)* deal with the task of land cover classification by employing the DeepLabV3 CNN *Chen et al. (2017b)* to extract the reflectance characteristics of snow cover in multispectral remote sensing data captured by the GF-2 satellite. The methodology presented in this work makes use of a pre-trained deep learning model with knowledge acquired from Landsat 8 training data. This knowledge was then specialized for the detection of snow cover using a manually labeled dataset composed of digital orthophoto maps (DOMs). The results of the experiment indicate that the framework is effective in automatically extracting information about snow cover. Deep learning approaches, such as fully-convolutional networks (FCN) (*Long, Shelhamer & Darrell, 2015*), U-NET (*Ronneberger, Fischer & Brox, 2015*), DeepLabV3 and dynamic dilated convolutional networks (DDCN) *Nogueira, Penatti & dos Santos, 2017* are widely employed for land-cover segmentation tasks, as seen in recent literature (*Balaniuk, Isupova & Reece, 2020*; *Nogueira et al., 2020*; *Bragagnolo, Da Silva & Grzybowski, 2021*; *Osco et al., 2021*). However, deep-based approaches often require larger and richer datasets for optimal performance, as highlighted by *Behera, Bakshi & Sa (2023)*.

Traditional machine learning algorithms, such as RF, offer reliable performance for pixel-wise segmentation tasks, as demonstrated by *Wu et al. (2019)*, *Linhui, Weipeng & Huihui (2021)*, *Conceição et al. (2021)*, *Luo et al. (2022)*, *Breiman (2001)*, particularly when dataset size and scope are limited. In remote sensing applications, including multispectral UAV image analysis, RF has shown results comparable to deep learning methods under constrained data conditions (*Behera, Bakshi & Sa, 2023*). Table 1 includes a summary of recent research applied to land-cover segmentation using both traditional and deep approaches.

There is considerable potential for pattern extraction in multispectral remote sensing images. However, the use of land cover assessment techniques in the context of DHM remains severely limited. This study intends to explore potential methods to increase the adoption of these techniques. The objective is to provide insight into the challenges associated with this technology and offer recommendations for effective implementation in the future.

## MATERIALS AND METHODS

In this work, a case study approach was adopted to evaluate the effectiveness of the RF algorithm for the land cover classification of embankment dams using multispectral remote sensing data. The complete methodology, as shown in Fig. 2, consists of three main stages: (1) Data Preparation, (2) Feature Engineering and (3) Model Training. Details of each step are presented in the following subsections.

**Table 1 Summarization of related literature's research problems, techniques and performance metrics.**

| Reference | Research problem | Techniques | Performance metrics |
|---|---|---|---|
| *Wu et al. (2019)* | Pixel-wise land-cover segmentation of urban areas using satellite and lidar data | Random forests | Accuracy: 94.51% |
| | | | Kappa: 93.0% |
| *Balaniuk, Isupova & Reece (2020)* | Mining and tailing dam detection in satellite imagery | Convolutional Neural Networks (FCN) | Accuracy: 90.42% |
| | | | Kappa: 85.50% |
| *Guo et al. (2020)* | Land-cover segmentation of snow areas using satellite imagery | Convolutional Neural Networks (DeepLabV3) | Pixel Accuracy: 91.0% |
| | | | IoU: 91.5% |
| *Nogueira et al. (2020)* | Pixel-wise deep-based erosion segmentation around railway lines using satellite data | Convolutional Neural Networks (DDCN) | Accuracy: 88.65% |
| | | | Kappa: 63.11% |
| | | | IoU: 53.55% |
| *Linhui, Weipeng & Huihui (2021)* | Pixel-wise land-cover segmentation of forest types using satellite data | Random Forests | Accuracy: 83.16% |
| | | | Kappa: 79.86% |
| *Conceição et al. (2021)* | Pixel-wise segmentation of oil spill in water surfaces using satellite data | Random Forests | Accuracy: 90% |
| *Bragagnolo, Da Silva & Grzybowski (2021)* | Binary image segmentation for forest cover change mapping using satellite imagery | Convolutional Neural Networks (U-NET) | Accuracy: 94.69% |
| | | | F-score: 94.69% |
| *Osco et al. (2021)* | Land-cover segmentation of citrus orchards using UAV multispectral data | Convolutional Neural Networks (DDCN) | Accuracy: 95.46% |
| | | | F-score: 94.42% |
| *Luo et al. (2022)* | Land-cover segmentation of mining areas using visible-band UAV imagery | Random Forests | Accuracy: 97.6% |
| | | | Kappa: 96.5% |
| *Behera, Bakshi & Sa (2023)* | Land-cover segmentation of vegetation areas | Convolutional Neural Networks (AerialSegNet) | Accuracy: 95.0% |
| | | | F-score: 82.0% |
| | | | IoU: 73.9% |

## Data collection and pre-processing

The Belo Monte Hydroelectric Complex is located in the northern part of the Xingu River, in the southwest region of the state of Pará, Brazil. The reservoir span a total area of 478 square kilometers and have a total installed capacity of 11,233.1 $MW$ (*NESA, 2020*). The study area of this work comprises part of the embankment dams and dikes located in the complex.

Autonomous UAV flight missions were executed at select dams and dikes in order to acquire multispectral images and compose the dataset used to train the machine learning models proposed in this study. The UAV system and multispectral camera sensor used in this task are shown in Fig. 3.

The DJI Matrice 210 V2 *DJI (2020)* is an enterprise-level UAV designed for commercial and industrial tasks, such as site mapping, infrastructure surveying, and construction inspection. The aircraft incorporates a dual battery system, providing it with a maximum flight time of 38 min. The dual-gimbal system supports a wide range of specialized payloads, including RGB, multi and hyperspectral cameras, and LiDAR sensors. The native mission planning mode embedded in the UAV's software allows for automatic planning and execution of image collection flights.

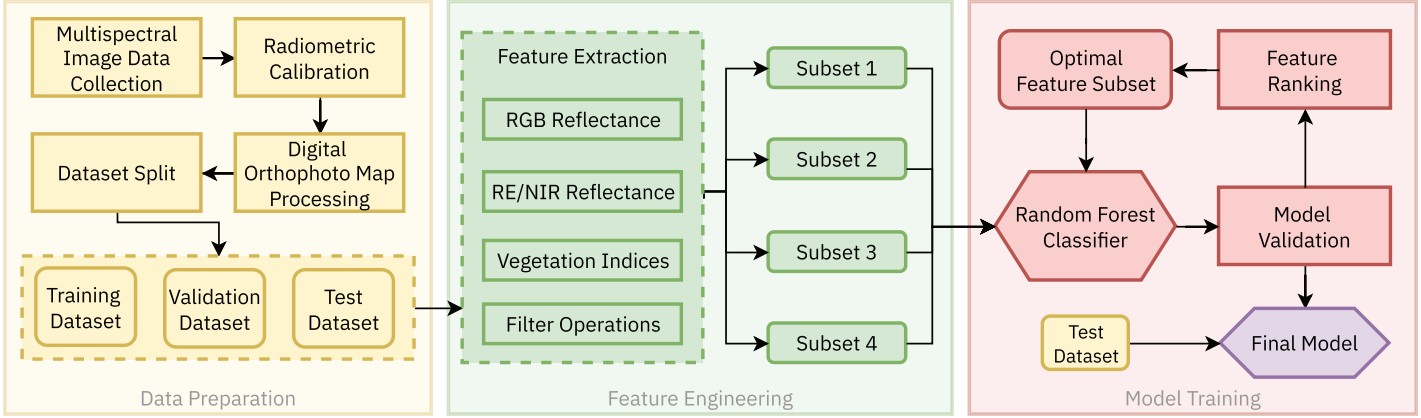

**Figure 2 The proposed methodology for land cover segmentation of earth-rock dams and dykes based on multispectral UAV imagery consists of three steps: (1) Data preparation, (2) Feature engineering and (3) Model training.**

Embedded to the DJI M210 V2 is the Micasense RedEdge-P multispectral camera (*MICASENSE, 2021*). Integration of this third-party sensor is facilitated by the included DJI SkyPort integration kit. The RedEdge-P is equipped with six optical sensors capable of capturing data from six distinct bands of the electromagnetic spectrum: red (R), green (G), blue (B), red edge (RE), near infrared (NIR), and panchromatic. Each sensor captures single-band images with a resolution of 1,456 × 1,088 pixels, which corresponds to 1.58 megapixels per band with a pixel size of 3.45 $\mu$. Table 2 summarizes the characteristics of the multispectral bands captured by the sensor.

The UAV flight plan was created beforehand using the built-in DJI mission planning module. Image collection campaigns were held on 16 and 17 May 2022. According to the manufacturer's instructions (*MICASENSE, 2021*), flight missions were carried out during sunny weather around 10 am in order to avoid the presence of shadows that can negatively affect the calculation of vegetation indexes. A fixed flight height of 80 m and a flight speed of 5 m/s were chosen to ensure an overlap of 75% between consecutive images.

A picture of the RedEdge-P calibration panel was taken from a height of approximately 1 m before the execution of each flight mission. This process allows for the computation of reflectance values during the radiometric calibration process. Reflectance is the ratio between the luminous flux reflected by the object and the luminous flux on the object (*Louargant et al., 2017*). The calibration process is possible since the reflectance values of the calibration panel surface are previously known. Due to the nature of this project, only open source software and tools were used in the development of this work. Radiometric calibrations as well as the band alignment process were performed using the Micasense Image Processing, provided by the sensor manufacturer. Radiometric calibration scripts are available on GitHub (*Micasense, 2024*). Panchromatic band data were incorporated in the pre-processing pipeline in order to enhance spatial resolution and feature visibility while preserving spectral information (*Garzelli et al., 2004*).

Image collection flights were performed on two different dam structures of the Belo Monte Complex, resulting in a total of 3,396 raw image files, corresponding to

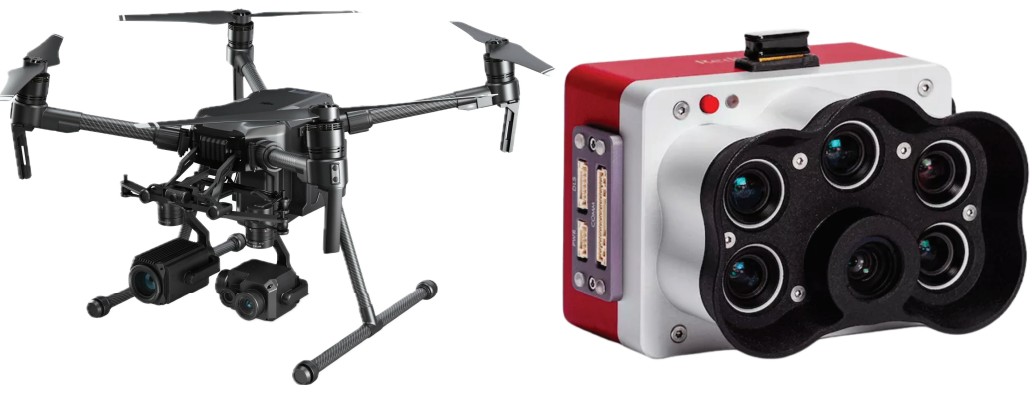

DJI Matrice M210 V2          Micasense RedEdge-P

**Figure 3 Automatic data collection missions were performed in a set of dams and dykes located in the study area.** The multispectral sensor Micasense RedEdge-P was mounted to the DJI Matrice 210 V2 UAV. 

**Table 2 Bands captured by the Micasense RedEdge-P multispectral sensor.**

| Band name | Central wavelength | Bandwidth |
|---|---|---|
| Blue (B) | 475 *nm* | 32 *nm* |
| Green (G) | 560 *nm* | 27 *nm* |
| Red (R) | 668 *nm* | 16 *nm* |
| Red-edge (RE) | 717 *nm* | 12 *nm* |
| Near infrared (NIR) | 842 *nm* | 57 *nm* |
| Panchromatic | 634.5 *nm* | 463 *nm* |

566 captures. The images were stitched together using the open source software OpenDroneMap (*OpenDroneMap, 2020*) in order to produce a DOM of the structures, which are shown in Fig. 4. The resulting DOMs are stored in the .tiff format and can be represented as a tensor of six channels, from which five stores reflectance data for each band and the last channel is added containing the binary cutline mask.

The data annotation process described in the following section was adopted to prepare the data for supervised semantic segmentation.

## Data annotation

Pixel-based data annotation for semantic segmentation problems is a notoriously complex task. This complexity contributes to the scarcity of high-quality data sets available to solve many specialized problems. Given the lack of pre-annotated aerial image data for semantic segmentation of embankment dam structures, the complete DOMs were manually annotated using the CVAT software (*Sekachev et al., 2020*). In this pixel-based segmentation task, each pixel of the input image must be assigned to a specific class. The definition of segmentation classes and the accuracy of the annotated labels were validated through continuous consultation with two domain specialists from the enterprise's dam safety team. The following four classes were chosen for classification in this study:

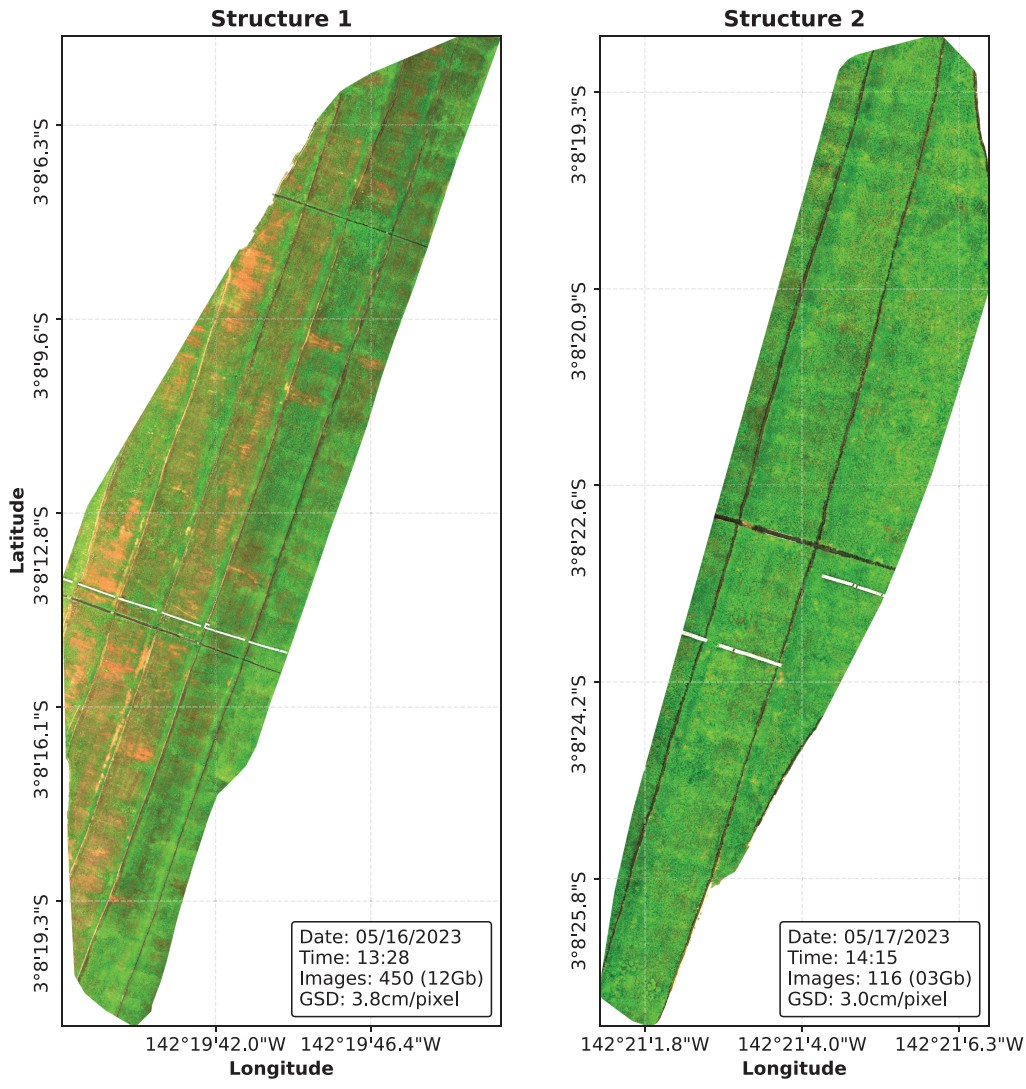

**Figure 4** **The Digital Orthophoto Maps (DOMs) were processed using the OpenDroneMap software.**

- **Slope:** Downstream slope of the embankment dam. In this work, this class represents the slope surface covered with vegetation.
- **Stairs:** Steel stairs that go from the bottom to the top of the structure.
- **Drainage channels:** Drainage structures found in each layer (berm) of the downstream slope.
- **Background:** Image background.

In order to create the dataset for pixel-based semantic segmentation, each pixel of the DOMs is represented as a row in tabular form, where the columns represent the corresponding pixels features. The dataset initially consists of seven features: grayscale pixels, reflectance values for each band captured by the RedEdge-P sensor, and the pixel

labels defined in this section. Each DOM was sliced into segments of 256 × 256 pixels, from which 70% was assigned to the training set, 15% to the validation set and 15% to the test set. Due to class imbalance in the training dataset, the majority classes (background and slope) were downsampled and the minority class (stairs) was upsampled in order to match the number of samples belonging to the Drainage Channels class. The original class distributions were kept for the validation and test sets. The total number of samples obtained for each class after the labeling process is summarized in Table 3. As an additional experiment, the multiclass dataset was adapted to perform binary segmentation as well, where all the samples belonging to classes other than Slope are considered as Not-Slope. A repository containing the complete dataset was published at the HuggingFace platform and is available at https://huggingface.co/datasets/andrematte/dam-segmentation (*Teixeira, 2024b*).

The next section describes the feature engineering strategies used to extract meaningful information that can be used to improve model performance.

## Feature engineering and selection

Feature engineering is a critical step in the machine learning pipeline, where additional sets of features are computed to improve the separability between the classes and thus improve model performance. In this work, the selected additional features can be divided into five groups: RGB Reflectance, Multispectral Reflectance, Vegetation Indices, Filtering Operations and Texture Features. A large group of features is selected at first, then a selection procedure based on feature importance ranking is applied to further optimize the dataset by ranking the most relevant features based on feature importance metrics. Examples of the extracted features are shown in Fig. 5.

The initial features are based on the calibrated reflectance measurements captured by the multispectral sensor. The different bands captured by the sensor are divided into two groups: the first group containing the RGB reflectance values, *i.e.*, data from the visible band, and the second group containing multispectral reflectance values, *i.e.*, data from the non-visible RE and NIR bands. This distinction is made in order to evaluate the impact of multispectral RS data in model performance.

The third group of features consists of various multispectral vegetation indices (VI), which are based on the absorption and scattering of electromagnetic radiation by vegetation (*Tsouros, Bibi & Sarigiannidis, 2019*). VIs are widely used in the field of RS and can combine RGB information with other spectral bands such as the NIR and RE. Features such as biomass, nitrogen status, water index, and vegetation health can be obtained by calculating different VIs. In this work, the following VIs were selected as features: Normalized Difference Vegetation Index (NDVI), Green NDVI (GNDVI), Normalized Difference Red Edge Index (NDRE), Green Chlorophyll Index (GCI), and Normalized Difference Water Index (NDWI).

The fourth set of features were extracted by filtering operations in order to highlight different characteristics of the input images. Six distinct filters were used for edge detection: Canny, Laplacian, Roberts, Sobel, Scharr, and Prewitt. Five features consists of

**Table 3 Number of samples (pixels) annotated for each class of land cover.** The dataset was divided in a stratified manner based on the hold-out strategy, with 70% of the samples assigned to the training set, 15% to the validation set and 15% to the test set.

| Class | Train samples | Validation samples | Test samples | Total |
|---|---|---|---|---|
| Slope | 707,843 | 1,404,283 | 1,835,333 | 3,947,459 |
| Drainage channels | 703,870 | 128,464 | 139,307 | 971,641 |
| Stairs | 707,843 | 33,676 | 40,205 | 781,724 |
| Background | 707,843 | 334,121 | 475,523 | 1,517,487 |

Gaussian and Median blurring operations with varying kernel sizes. Finally, a series of 64 Gabor filters with different parameters was generated and applied to the input images.

A feature subset based approach was used to select the most meaningful subsets of features to be used in the final model training process. The non-visible bands captured by the multispectral sensor were purposefully omitted from some of the subsets in order to assess the contribution of the RE and NIR bands to model performance. The following six subsets were chosen for the experiments (the grayscale image pixels are included in all subsets):

- **Subset 1: RGB reflectance (four features)**—Includes reflectance data captured on visible bands (R, G, and B). This scenario emulates the use of traditional camera sensors;
- **Subset 2: Multispectral reflectance + VI (11 features)**—Includes the five bands captured by the multispectral sensor, including the non-visible ones (R, G, B, Red Edge and NIR) and adds the resulting Vegetation Indices;
- **Subset 3: RGB + filters (21 features)**—Includes the RGB bands and all the features extracted by the filtering operations;
- **Subset 4: Multispectral + VI + filters (27 features)**—Includes all the bands captured by the sensor, the VIs and all the features extracted by the filtering operations;

## Semantic image segmentation

In the computer vision and remote sensing fields, the goal of semantic segmentation problems is to assign a class to each individual pixel of the input image. This process divides the image into visually meaningful or interesting areas, allowing subsequent image analysis and visual understanding (*Mo et al., 2022*). The learning-based approach of machine learning models improved the performance of classic semantic segmentation approaches (*Kotaridis & Lazaridou, 2021*). A pool of well established machine learning techniques, such as KNN, SVM, and RF, are capable of performing well on semantic segmentation tasks. Particularly, the RF algorithm has demonstrated strong performance in semantic segmentation tasks due to its capacity to handle high-dimensional datasets, adaptability to different features types, interpretability, suitability to handle imbalanced data classes and resistance to overfitting. RF has consistently shown satisfactory results in the literature when applied to image segmentation (*Linhui, Weipeng & Huihui, 2021*;

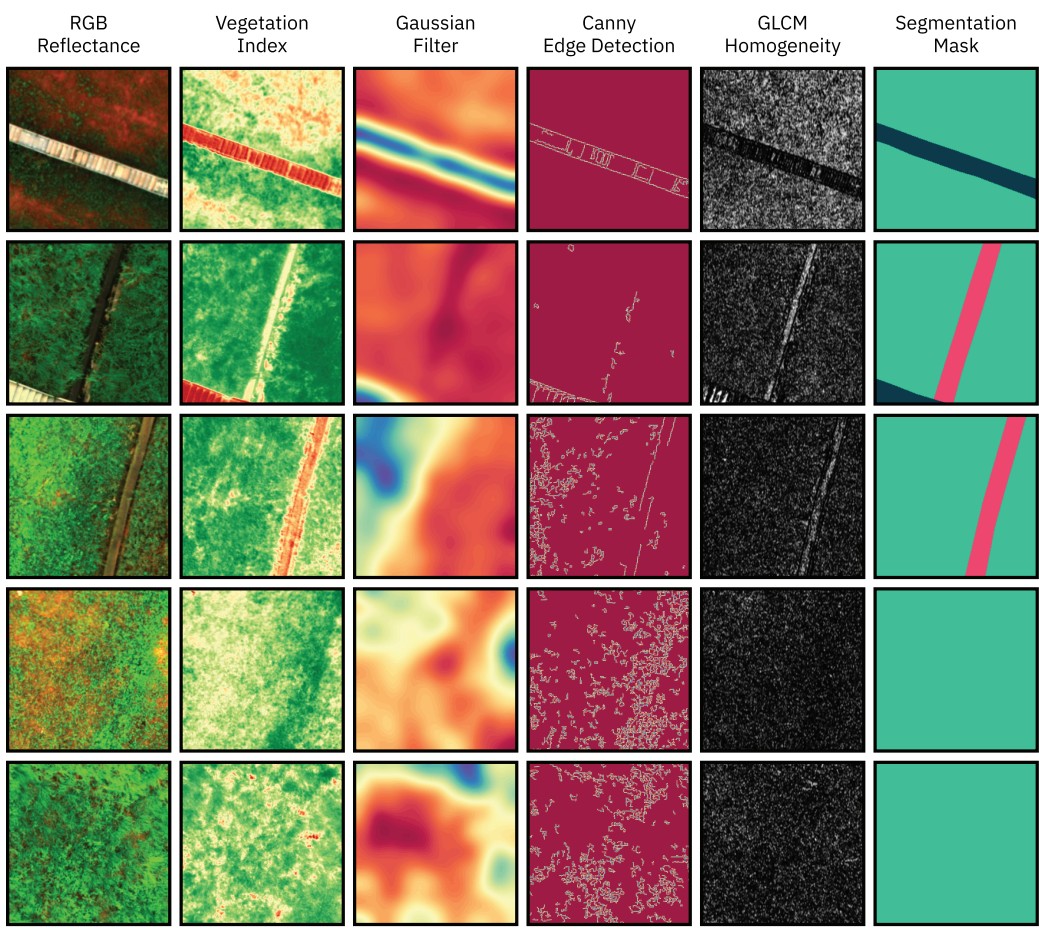

**Figure 5 Examples images from the dataset (RGB representation), manually extracted features, and manually annotated ground truth segmentation masks.**

*Conceição et al., 2021*; *Wu et al., 2019*), and thus was selected as the pixel-wise classifier for the land-cover segmentation problem addressed in this work.

The RF classifier is a machine learning technique based on an ensemble of individual decision tree (*Quinlan, 1986*) models, as depicted in Fig. 6. Each model in this collection of tree-structured classifiers, $h(x, \Theta_k), k = 1, \ldots$ where the $\theta_k$ are independent identically distributed random vectors (*Breiman, 2001*). As an ensemble model that consists of different classifiers and given an input $X$, each decision tree casts a vote for the most popular class, contributing to the definition of the output $y$. The advantages of the RF algorithm include being nonparametric, not requiring assumptions on the distribution of training data, being capable of running efficiently with large datasets and being able to rank feature importance (*Wu et al., 2019*).

## Model performance assessment

Model performance metrics based on the confusion matrix were chosen to evaluate the models. Accuracy assessment for classification problems is usually achieved by comparing

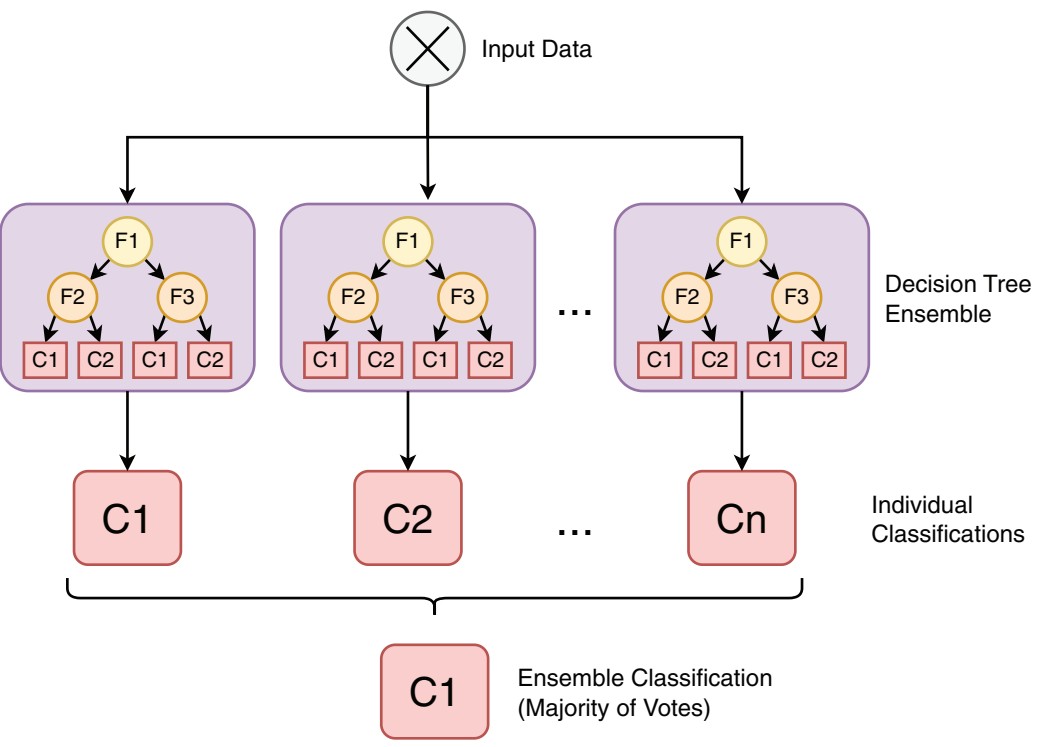

**Figure 6 Random forest is an ensemble classification algorithm composed of individual decision tree models.**

the data points classified by an algorithm with previously known data points of the test set (*Lewis & Brown, 2001*). Then, these comparison data are summarized in the form of a confusion matrix (Table 4). The main diagonal of the matrix represents the correct classified samples, *i.e.*, the true positives (TP) and true negatives (TN). The remaining cells represent the incorrectly classified samples, *i.e.*, the false positives (FP) and false negatives (FP). This comparison format is easily generalized for problems with more than two classes.

Various descriptive and analytical measures based on the confusion matrix can be used to summarize the accuracy of a classification model (*Lewis & Brown, 2001*). The metrics used in this work are accuracy, Kappa coefficient, precision, recall, and F1 score.

$$\text{Accuracy} = \frac{TP + TN}{P + N} = \frac{CorrectlyPredicted}{Total}. \tag{1}$$

The accuracy metric (Eq. (1)) is used to evaluate the correctness of the output of a classification model. It can be achieved by calculating the ratio between the correct classified samples ($TP + TN$) and the total number of classified samples ($P + N$). Although higher accuracy values might indicate that the model is performing well, this metric is not suitable for unbalanced data sets. Therefore, other evaluation metrics, such as precision and recall, are also considered.

**Table 4 Example confusion matrix for a 2-class classification problem** *Rácz, Bajusz & Héberger (2019).*

|  | Predicted positive (PP) | Predicted negative (PN) |
|---|---|---|
| Actual Positive (P) | True Positive (TP) | False Negative (FN) |
| Actual Negative (N) | False Positive (FP) | True Negative (TN) |

$$\text{Precision} = \frac{TP}{TP + FP} \tag{2}$$

$$\text{Recall} = \frac{TP}{P} = \frac{TP}{TP + FN}. \tag{3}$$

A comprehensive evaluation of model performance must include both precision (Eq. (2)) and recall (Eq. (3)), which are often in conflict. Precision, also known as positive predictive value, is used to assess the quality of positive class classifications. High precision implies a reduced false positive rate. This metric can be very significant in scenarios where detecting false positives samples can lead to higher costs or inaccuracies. On the other hand, the recall metric, often called sensitivity or true positive rate, is used to calculate the proportion of actual positive samples that were correctly classified. This metric can be particularly significant in scenarios such as fraud detection or medical analysis, as missing a positive instance can have severe consequences.

$$\text{F1 score} = 2 * \frac{Precision * Recall}{Precision + Recall} = \frac{2 * TP}{2 * TP + FP + FN}. \tag{4}$$

Precision and recall must be evaluated together due to the conflict between both metrics. The F1 score (Eq. (4)) is the harmonic mean between precision and recall, and is used to summarize the performance of the model.

Cohen's Kappa Incluir K

$$\text{Cohen's Kappa} = \frac{P(A) - P(E)}{1 - P(E)}. \tag{5}$$

The Cohen Kappa score (*Ferri, Hernández-Orallo & Modroiu, 2009*), or Kappa coefficient (Eq. (5)), is based on the confusion matrix and is often used to compare the effectiveness of different machine learning models while helping to account for potential biases and random variations. This metric takes into account the observed accuracy $P(A)$ of the model and compares it with an expected accuracy value that could be achieved with a random classifier, denoted by $P(E)$.

$$\text{Mean IoU} = \frac{1}{c} \sum_i \frac{n_{ii}}{t_i + \sum_j n_{ji} - n_{ii}}. \tag{6}$$

With the advancement of the field of computer vision, tasks involving pattern extraction from image data have required the creation of specific metrics for comparing images. The intersection over union (IoU) is a metric that measures the overlap between images or segmentation masks generated by the model and the actual data. The metric is calculated as the ratio between the intersection and union of the images or masks, providing a

measure of similarity between them (*Long, Shelhamer & Darrell, 2015*). The mean IoU metric is defined by Eq. (6), where $n_{i,j}$ denotes the number of pixels belonging to class $i$ that were classified as belonging to class $j$, $t_i$ represents the total amount of pixels belonging to class $i$ and $c$ represents the number of classes.

## RESULTS AND DISCUSSION

### Feature selection

In this first model training step, a grid search approach is performed to train multiple RF models with varying numbers of classifiers, represented by the parameter *nTrees*, in order to determine the optimal configuration. Each of the four subsets was trained and evaluated for ensembles of 2, 4, 8, 16, 32, 64, 128 and 256 decision trees. Performance assessment was conducted using the validation dataset based on the macro F1 score, OOB error and elapsed training time. A total of 64 unique models were trained and evaluated during this stage, and their respective performance metrics are presented in Fig. 7. All experiments in this work were performed on a computer with an 8-core Apple Silicon M1 Pro processor running at 3.22 GHz, 16 GB of RAM, running the macOS Sequoia 15.1 operating system. The complete source code is available at GitHub (https://github.com/andrematte/dam-segmentation) (*Teixeira, 2024a*). The best results obtained for each set of features and parameters are summarized in Table 5.

In contrast to Subset 1, which was trained solely on RGB visible reflectance data, the insertion of the RE and NIR reflectance data captured by the multispectral sensor and VIs in Subset 2 has led to a contribution of up to 2.46% and 6.98% in macro F1 score for the binary and multiclass problems, respectively. Models based on the reflectance values captured by the RedEdge-P multispectral sensor and VIs achieved overall accuracies of up to 97.19% and 89.73% without further feature engineering.

The third subset is composed of RGB reflectance data and features generated by filtering operations. Reflectance data for the non-visible bands (RE and NIR) are purposefully omitted in these scenarios for comparison purposes and to assess the impact of the multispectral sensor data in model performance. Compared to Subset 1, macro F1-score increased from 94.73% to 95.41% for binary segmentation and from 82.75% to 88.68% for the multiclass task. This experiment aims to simulate situations where multispectral sensor equipment or data are unavailable. In such scenarios, applying the feature engineering process to the dataset can lead to a significantly improved accuracy in the multiclass classification task.

In Subset 4, the combination of all the proposed features, including RGB and Multispectral reflectance, VIs, Filtering Operations and texture features resulted in the best measured performance. Feature Subset 4 has achieved 97.02% balanced accuracy and 97.46% macro F1 score for the binary segmentation. As for the multiclass segmentation task, 96.58% balanced accuracy and 92.19% macro F1 score were attained. All the best ensembles of each feature subset are composed of 256 individual decision tree models.

Although the performance of models trained solely on RGB reflectance data tends to be lower, the large amount of features that can be generated by operations on these bands is often enough to achieve high accuracies on pixel classification tasks such as land cover

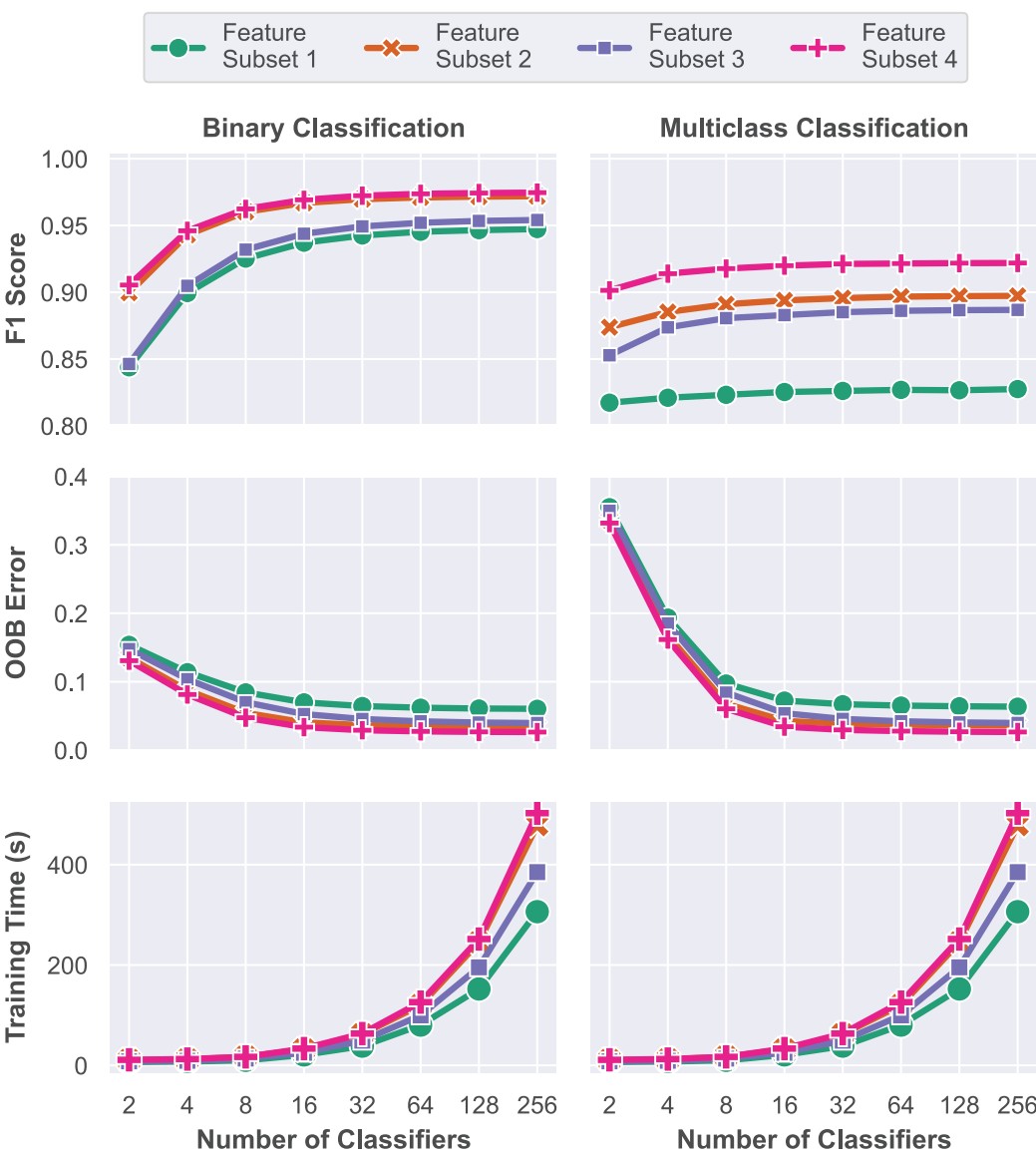

**Figure 7** Model performance evaluation based on overall accuracy, Cohen's Kappa coefficient and training time for each model explored during the grid search methodology.

segmentation. Operations such as image smoothing by blurring filters, edge detection, CNN feature extractors and texture extractors are capable of producing high-quality features that highlight different characteristics in the input images. However, several spectral characteristics are not captured by traditional RGB sensors, leaving room for improvement with the addition of non-visible band reflectance data.

## Feature ranking

Training a RF classifier with an exceedingly large set of features might lead to overfitting (*Linhui, Weipeng & Huihui, 2021*). Thus, in this step, a feature ranking methodology was

**Table 5 Model performance metrics, optimal number of features and optimizers (ntrees) for each subset of features.** Balanced Accuracy and Macro F1 score metrics are calculated to account for the class imbalance in the test set. The best combination of parameters were selected and highlighted in bold.

| Task type | Subset | Features | nTrees | Accuracy | F1 score | OOB error | Training time |
|---|---|---|---|---|---|---|---|
| Binary | Subset 1 | 4 | 256 | 93.98% | 94.73% | $6.04 * 10^{-2}$ | 306 s |
| Binary | Subset 2 | 10 | 256 | 96.73% | 97.19% | $3.40 * 10^{-2}$ | 479 s |
| Binary | Subset 3 | 63 | 256 | 94.98% | 95.41% | $3.96 * 10^{-2}$ | 385 s |
| **Binary** | **Subset 4** | **69** | **256** | **97.02%** | **97.46%** | $2.64 * 10^{-2}$ | **502 s** |
| Multiclass | Subset 1 | 4 | 256 | 93.05% | 82.75% | $6.35 * 10^{-2}$ | 330 s |
| Multiclass | Subset 2 | 10 | 256 | 95.89% | 89.73% | $3.62 * 10^{-2}$ | 543 s |
| Multiclass | Subset 3 | 63 | 256 | 95.06% | 88.68% | $3.97 * 10^{-2}$ | 410 s |
| **Multiclass** | **Subset 4** | **69** | **256** | **96.58%** | **92.19%** | $2.65 * 10^{-2}$ | **581 s** |

applied in order to select and keep only features with significant contributions to overall model performance. The importance of each feature was calculated on the basis of the mean decrease in impurity (MDI) metric (*Scornet, 2021*). A clear upward trend was observed as features were sequentially added based on their importance rankings. With the sequential addition of features following the order of contribution rates, there is a steep upward trend in performance, peaking at 97.42% F1 score for binary segmentation and 92.15% for multiclass segmentation. Model performance metrics for each problem and the feature importance ranking results are shown in Fig. 8.

The feature importance rankings reveal that vegetation indices (NDWI, GNDVI, NDVI, NDRE) and spectral bands, particularly in the visible and near-infrared range, were the most influential in both segmentation tasks. Notably, the Blue band stood out as a strong predictor. Additionally, although spatial texture features derived from Gaussian, Median, and Gabor filters had lower individual importance scores, their consistent presence among the top features suggests that local texture plays an important complementary role in distinguishing between structural elements like slopes, stairs, and drainage channels. Overall, the results demonstrate that a compact and well-ranked feature set is sufficient to achieve high segmentation performance while avoiding redundancy and reducing computational cost.

## Model evaluation

The final model for the binary segmentation task was trained using the 17 features that contributed the most to its performance. The multiclass model was trained with the 24 most significant features. These models were evaluated on the hold-out test dataset according to metrics derived from the confusion matrix presented in Fig. 9. Based on the metrics presented below, it is possible to assess the quality of the trained model and its ability to differentiate between classes.

Metrics such as precision, sensitivity, F1 score, and IoU were calculated for each class based on the confusion matrix and are summarized in Table 6. Additionally, the overall metrics of the final models are also presented.

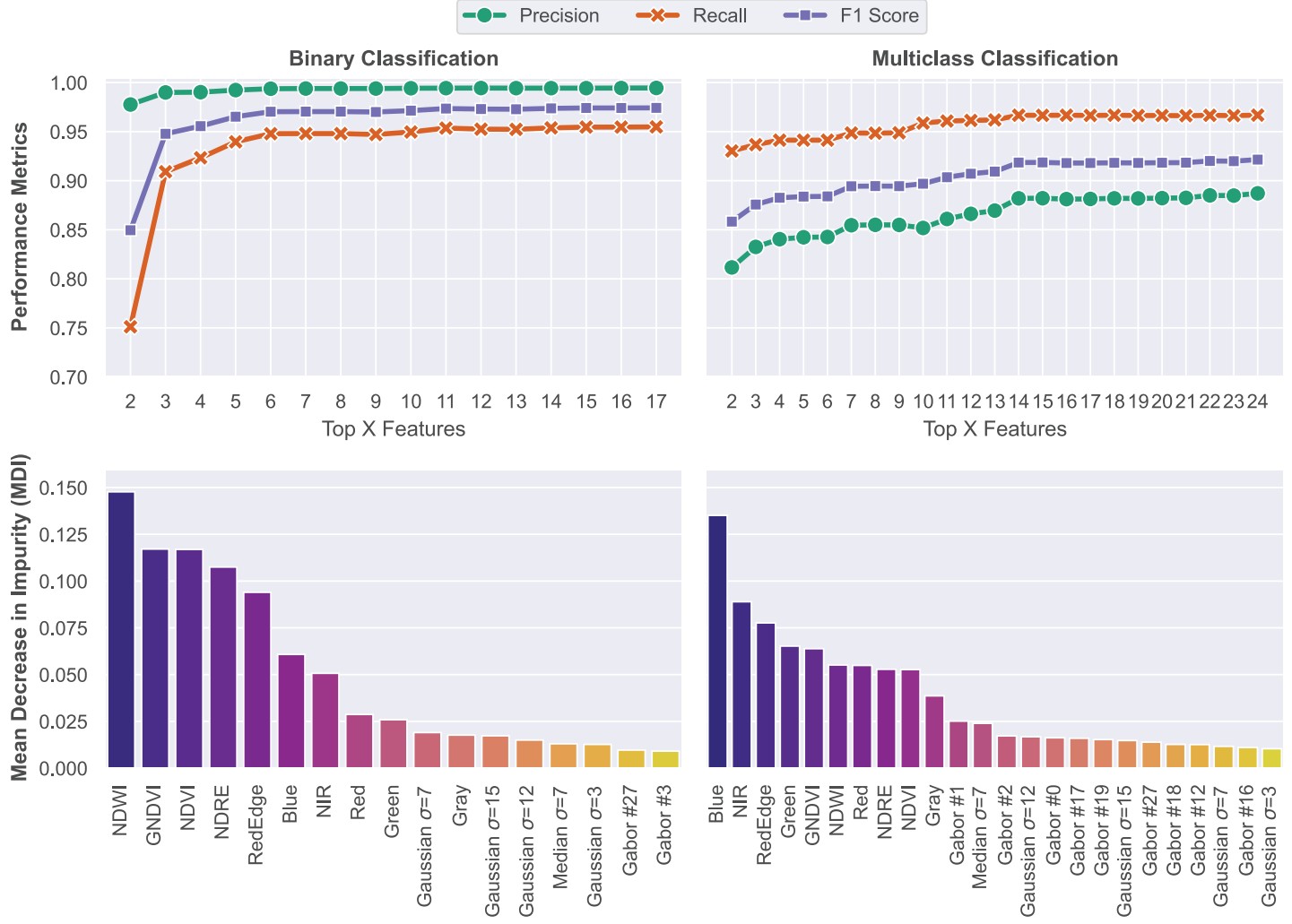

**Figure 8** **Feature ranking methodology based on the MDI of each feature.** Models were trained on features that achieved MDI values higher than 0.01.

The results obtained for the binary segmentation task reached 98.4% F1-score and 94.8% IoU for the classification of slope surface pixels. However, for the negative class, which represents not only the background but also the classes of stairs and drains, the metric dropped to 87.0%. This drop in both metrics indicates an increase in the false positive rate, meaning that pixels that should have been classified as Not-Slope were classified as Slope. Despite the lower precision of 88.2%, the model's sensitivity for the negative class remained at 95.3%. Overall, the binary classifier achieved 97.3% in F1-score and 90.9% in IoU, suggesting a consistent performance when all the classes are combined.

The multiclass segmentation task achieved satisfactory F1-score levels above 97.9% and IoU levels above 95.8% for all the study's classes except for the drainage channels, which achieved 82.9% F1-score and 71.2% IoU. Higher metrics were expected for the background

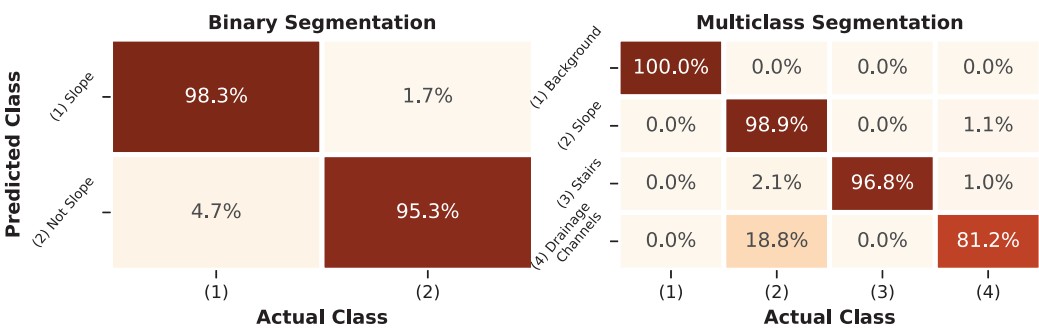

**Figure 9 Confusion matrix of the RF segmentation model.** The percentage values present on the main diagonal of the matrix represent the correctly classified instances for each class in the test dataset.

**Table 6 Class-specific and overall performance metrics for the binary and multiclass segmentation models.**

**Binary classification**

| Class | Precision | Recall | F1-score | IoU |
|---|---|---|---|---|
| Slope | 0.994 | 0.953 | 0.973 | 0.948 |
| Non slope | 0.882 | 0.984 | 0.930 | 0.870 |

**Multiclass classification**

| Class | Precision | Recall | F1-score | IoU |
|---|---|---|---|---|
| Background | 0.999 | 0.999 | 0.999 | 0.999 |
| Slope | 0.985 | 0.989 | 0.987 | 0.975 |
| Stairs | 0.990 | 0.968 | 0.979 | 0.958 |
| Drainage channels | 0.850 | 0.814 | 0.832 | 0.712 |

**Overall performance metrics**

| Class | Binary | Multiclass |
|---|---|---|
| Kappa coefficient | 0.9032 | 0.9534 |
| Balanced accuracy | 0.9682 | 0.9422 |
| Precision | 0.9938 | 0.9554 |
| Recall | 0.9531 | 0.9422 |
| F1-score | 0.9730 | 0.9487 |
| Mean IoU | 0.909 | 0.911 |
| Training time | 500s | 2061s |

class, as these pixels contain no information. The lowest performance of the drainage channels class can be attributed to the quality of the image labeling process when it comes to fine details of the transitions between the slope and the drainage channels of the structures. The confusion matrix indicates that the slope class was correctly classified 98.9% of the time, but the drainage channels class was confused with the slope class 18.8% of the time, which harmed its precision metric.

## DISCUSSION

The performance metrics obtained by the RF model were promising for both binary and multiclass segmentation tasks, demonstrating the viability of the proposed method and enabling further analysis of the segmented regions. Given that the multiclass approach offers a more detailed breakdown of the image regions, it is the preferred option in this context. A comparison between ground truth masks, predictions generated by the best multiclass model, and the map of incorrect classifications is shown in Fig. 10.

Although the overall performance of the model was satisfactory, the Drainage Channels class presented lower metrics. The main segmentation errors made by the model occurred at the boundaries between the Slope and the Drainage Channels classes, with the drainage channels performance metrics being the most affected. These errors can be attributed to several factors, such as: (1) overgrown vegetation on the slopes encroaching into the drainage channels, blurring the visual distinction between the two classes; and (2) annotation limitations, as the masks sometimes fail to capture overgrown vegetation with fine-grained precision.

Despite the satisfactory quantitative performance obtained from the evaluation of the presented performance metrics, the qualitative visual analysis of the resulting masks reveals the presence of scattered point noise sections representing different classes that are inconsistent with the pixel's neighborhood. The presence of noise negatively impacts the quality of segmentation, hindering the interpretability and reliability of the model's inferences.

The salt-and-pepper effect is a common phenomenon in pixel-by-pixel segmentation approaches that negatively impacts the quality of the resulting masks (*Paredes-Gómez et al., 2020*; *Csurka, Volpi & Chidlovskii, 2023*). The occurrence of this effect reduces the visual quality of the segmentation masks and directly affects the performance metrics used to evaluate the model. To mitigate this issue, it is advisable to incorporate features that consider the neighborhood of each pixel, facilitating context interpretation. It is also possible to mitigate the problem by applying post-processing filters to the resulting masks to remove noise and smooth the segmentation mask, improving the continuity of each segmented region (*Linhui, Weipeng & Huihui, 2021*).

To reduce isolated misclassifications caused by the salt-and-pepper and shadowing effects and improve spatial coherence in the predicted segmentation masks, majority filtering was applied as a post-processing step. Specifically, a sliding window of size $k \times k$ (with $k = 3$) was applied to each output image. Each pixel was reassigned to the most frequent class within its local neighborhood. Figure 10 presents a comparison between RGB images, ground truth segmentation masks, model predictions, and refined predictions.

The comparison between the annotated, predicted, and refined segmentation masks is made using the IoU metric, as shown in Fig. 10. The post-processing of predictions resulted in improvements in the average IoU across all images of the test set, along with a significant enhancement in the visual quality of predictions by removing salt-and-pepper

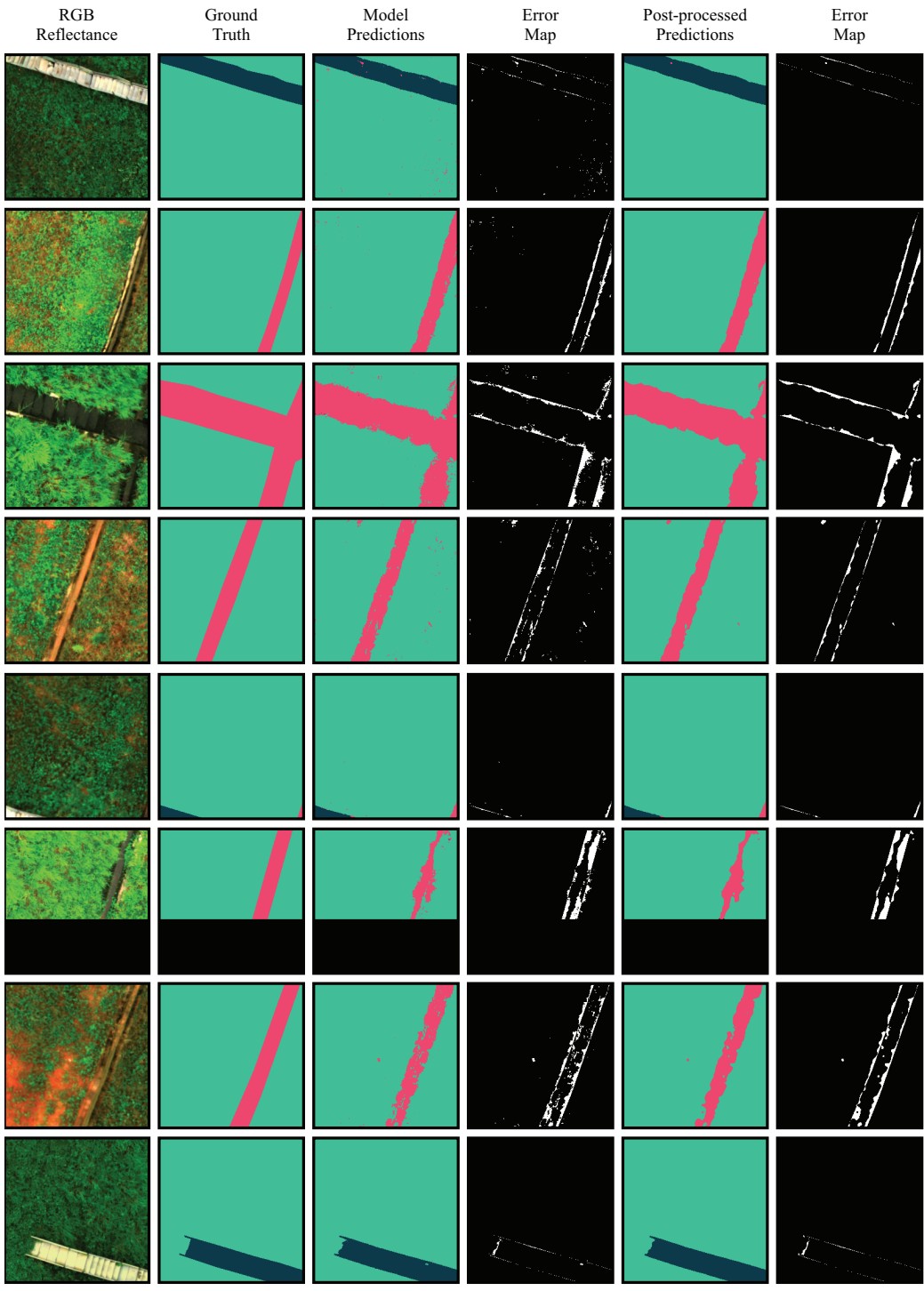

| RGB Reflectance | Ground Truth | Model Predictions | Error Map | Post-processed Predictions | Error Map |

**Figure 10 Comparison between RGB images, ground truth segmentation masks, model predictions and refined predictions.**

noise. To break down the effects of post-processing, Table 7 reports the IoU of the raw predictions and the refined outputs. In summary, the post-processing improved the segmentation masks IoU by up to 3.68%.

**Table 7 Comparison of class-specific and overall IoU between raw model predictions and post-processed segmentation masks.**

**Binary classification**

| Class | Raw IoU | Refined IoU |
|---|---|---|
| Slope | 0.948 | 0.962 (+1.48%) |
| Non slope | 0.870 | 0.902 (+3.68%) |

**Multiclass classification**

| Class | Raw IoU | Refined IoU |
|---|---|---|
| Background | 0.999 | 0.999 (+0.00%) |
| Slope | 0.975 | 0.977 (+0.21%) |
| Stairs | 0.958 | 0.963 (+0.52%) |
| Drainage channels | 0.712 | 0.738 (+3.65%) |

**Overall performance metrics**

| Metric | Binary | Multiclass |
|---|---|---|
| Raw IoU | 0.909 | 0.911 |
| Refined IoU | 0.932 (+2.53%) | 0.919 (+0.88%) |

Although the post-processing filters improved both the visual quality of the segmentation masks and the IoU scores, they do not fully capture the underlying spatial relationships between neighboring pixels. A more robust solution would involve the use of deep CNN models, such as FCN (*Long, Shelhamer & Darrell, 2015*), U-NET (*Ronneberger, Fischer & Brox, 2015*) or DeepLab (*Chen et al., 2017a*), which inherently account for pixel neighborhoods and contextual information. However, these models typically require larger amounts of labeled data for training, which can be a limitation in niche problems such as the segmentation of earth dams. Despite these challenges, the shift toward deep learning would likely yield better results, reducing noise and improving generalization beyond what traditional methods like RF can achieve.

## CONCLUSIONS

The objective of this study was to evaluate the applicability of machine learning techniques for land cover segmentation based on multispectral images as a step towards the automation of visual inspection routines. Data collection routines using multispectral sensors embedded in UAVs were carried out in structures present in the study area, which is located at the Belo Monte Hydroelectric Power Plant in the southwest region of the state of Pará, northern Brazil. A comprehensive feature extraction was conducted to create a robust dataset containing different types of features. The model was then optimized by a feature selection process, retaining only the most significant features, thus reducing computational time. The final model for binary segmentation was trained on the top 17 features and achieved a 93.2% IoU, while the multiclass model was trained on the top 24 features and achieved a satisfactory IoU of 91.9%.

The key findings of this article can be summarized as follows:

1. The distinct spectral signatures of individual types of land cover within the earth-rock dam structures of the study area enable precise pixel-level segmentation. Accurate segmentation will allow for different types of subsequent analyses related to structural integrity;

2. The experiments show that, while it is possible to achieve satisfactory accuracy levels by using only RGB images, the introduction of features from the multispectral bands is capable of significantly improving model accuracy;

3. The salt-and-pepper effect, caused by the lack of spatial context understanding when using traditional ML algorithms such as RF, can be mitigated by post-processing techniques such as image smoothing, improving the visual quality of the segmentation masks.

The use of UAVs for DHM presents key operational challenges that must be addressed for scalable deployments. Battery life remains a primary limitation, as most commercial UAVs operate for less than an hour per cycle (*Shakhatreh et al., 2019*). In this study, a full data collection mission for a single structure was completed within a single battery cycle. However, large-scale monitoring of multiple structures requires optimizing fleet size and backup batteries to maintain continuous data collection. Stability issues due to wind and variable weather conditions can be mitigated through real-time path planning and adaptive flight control, ensuring accurate georeferencing and image alignment.

This study represents a first step toward advancing image-based monitoring in the dam industry, addressing the critical gap caused by the lack of publicly available datasets. Without accessible data, the development of innovative machine and deep learning solutions remains limited. This work lays the groundwork for future advancements by establishing a foundational dataset. Expanding the dataset to include a more diverse range of structures and environmental conditions is a key priority for future work, enabling the development of deep learning models for image segmentation and anomaly detection, which require larger and more varied data for optimal performance.

Future research will also explore the feasibility of real-time monitoring, assessing how automated and continuous data collection methods can be integrated into the proposed workflow. While real-time analysis is not a current requirement for detecting gradual vegetation anomalies, advancing toward more frequent or automated assessments could enhance early warning capabilities in dam safety. Additionally, scalability assessment will be a critical focus, evaluating how this methodology can be effectively implemented across multiple large dams with varying geotechnical and environmental conditions. Further development will involve optimizing data acquisition and processing strategies to enable broader adoption of UAV-based monitoring frameworks in dam safety management. Furthermore, the framework presented in this work can be adapted for a broader range of geotechnical structures by expanding the dataset and tailoring the segmentation model to different environmental and geological contexts.

## ACKNOWLEDGEMENTS

We acknowledge the use of the ChatGPT tool for grammar checking and English language refinement.

### Funding

This work was financed by CAPES (FC 88887.650046/2021-00) under the ANEEL R&D Program (PD-07427-0321/2021) funded by Norte Energia S.A. There was no additional external funding received for this study. This work was developed as a R&D project funded by Norte Energia S.A. and the publication was approved by them.

### Grant Disclosures

The following grant information was disclosed by the authors:
CAPES: FC 88887.650046/2021-00.
ANEEL R&D Program: PD-07427-0321/2021.

### Competing Interests

Marcos A. Costantin Filho is an employee of Norte Energia S.A..

### Author Contributions

- Carlos André de Mattos Teixeira conceived and designed the experiments, performed the experiments, analyzed the data, performed the computation work, prepared figures and/or tables, authored or reviewed drafts of the article, and approved the final draft.
- Thabatta Moreira Alves de Araujo conceived and designed the experiments, authored or reviewed drafts of the article, and approved the final draft.
- Evelin Cardoso conceived and designed the experiments, authored or reviewed drafts of the article, and approved the final draft.
- Marcos Antonio Costantin Filho conceived and designed the experiments, authored or reviewed drafts of the article, and approved the final draft.
- João Weyl Costa conceived and designed the experiments, authored or reviewed drafts of the article, and approved the final draft.
- Carlos Renato Lisboa Frances conceived and designed the experiments, authored or reviewed drafts of the article, and approved the final draft.

### Data Availability

   The dataset is available at HuggingFace: https://doi.org/10.57967/hf/3089.
   The source code is available at GitHub and Zenodo:
   - https://github.com/andrematte/dam-segmentation/tree/v1.1.0
   - André Mattos. (2024). andrematte/dam-segmentation: v1.1.0 (v1.1.0). Zenodo. https://doi.org/10.5281/zenodo.13984238.

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
