# Peer review of "Aerial image segmentation of embankment dams based on multispectral remote sensing: a case study in the Belo Monte Hydroelectric Complex, Pará, Brazil"

_PeerJ Computer Science, doi:10.7717/peerj-cs.2917_

## Round 0.1 · original submission · Major Revisions

The manuscript explores the use of multispectral remote sensing and Random Forest (RF) for aerial image segmentation of embankment dams in the Belo Monte Hydroelectric Complex. While the study demonstrates innovation and potential real-world applications, reviewers highlighted several areas for improvement in its presentation, methodology, and findings:

Presentation and General Comments
- The manuscript lacks a performance comparison with deep learning models like U-Net or DeepLab, which are standard in segmentation tasks and may better address pixel-level spatial relationships.
- Limited dataset scope (two dam structures) raises questions about generalizability to diverse environmental and geological conditions.
- Manual segmentation reliability is unclear due to the absence of inter-annotator agreement analysis.
- The omission of the Panchromatic band and insufficient discussion of dataset limitations (e.g., weather variability, light conditions) weaken the methodology.

Experimental Design
- Potential overfitting during feature selection and the impact of adding/removing features (e.g., vegetation indices, filters) on performance need more analysis.
- The absence of the Intersection over Union (IoU) metric, a standard in segmentation evaluations, reduces the robustness of the findings.
- Scalability for large-scale or real-time dam monitoring systems remains unexplored.

Validity of Findings
- The study does not provide actionable recommendations for dam safety teams or a roadmap for operational implementation.
- Errors in segmenting drainage channels (F1-Score: 82.9%) highlight limitations in detecting fine details, and outlier analysis is missing.
- The relationship between segmentation tasks and dam collapse risk needs clearer framing to strengthen the study’s motivation.
- Additional visual comparisons (e.g., overlaying segmentation masks on input images) would enhance interpretability.

Key Suggestions
- Compare RF performance with deep learning models like U-Net and DeepLab to clarify trade-offs and benefits.
- Extend the dataset to include diverse dam structures and varying conditions (e.g., seasonal changes).
- Conduct a detailed analysis of segmentation errors and improve discussion on specific limitations (e.g., classifying drainage channels).
- Provide clear steps for integrating the methodology into dam monitoring workflows, addressing scalability and real-time applicability.

We invite the authors to carefully address all the comments raised before resubmission and provide a rebuttal letter with evidence of the main changes implemented in the manuscript.

Reviewer 1 ·

Basic reporting

This paper proposed the Aerial image segmentation of embankment dams based on Multispectral Remote Sensing. Overall, the structure of this paper is well organized. The presentation is clear to the readers. Some detailed comments can be found as follows.
1. Please clarify what is the current challenge compared to general environment.
2. What is the main contribution? What is the main difference compared to existing methods?

Experimental design

Some visual results can be added to provide a more comprehensive analysis in experiments.

Validity of the findings

Many important works on advanced machine/deep learning in classification and segmentation using remote sensing data are missing and need to be further discussed.

Reviewer 2 ·

Basic reporting

The main purpose of the article is to assess the classification performance of the algorithm in segmenting earth-rock dams and the contribution of non-visible band reflectance data to the overall model performance. It uses RF models that have been trained on datasets captured by UAVs. What are the unique features of the comparison with other latest models? It is recommended to add multiple model performance comparison experiments.
In terms of motivation, it is recommended to briefly introduce the relationship between segmentation and dam collapse at the beginning, as well as how segmentation can be used to determine the situation of dam collapse.

Experimental design

In Figure 1, the meaning of the location of the validation dataset in model training is unclear. It is recommended to modify this figure.
Why is the traditional image segmentation metric IoU not used as one of the indicators to measure model performance when using binary segmentation and multi category segmentation?

Validity of the findings

The impact and novelty of the study were not directly evaluated in the paper. Although it is mentioned that the purpose of this study is to promote the automation of visual inspection processes and the application of multispectral images in land cover segmentation is pointed out, there is no clear indication of the uniqueness of this study compared to existing research or its new contribution to the field.

Additional comments

The results analysis section is relatively detailed, but it is recommended that the author add an analysis of the outliers in the segmentation results in the discussion section, exploring possible causes and solutions.
Algorithm optimization: The author has made innovations in the algorithm, but there is still room for further optimization. Suggest the author to consider introducing more advanced machine learning algorithms or deep learning models to improve segmentation accuracy and efficiency.

Reviewer 3 ·

Basic reporting

see my comments

Experimental design

see my comments

Validity of the findings

see my comments

Additional comments

The article investigates the use of multispectral remote sensing and machine learning, specifically Random Forest, for aerial image segmentation of embankment dams in the Belo Monte Hydroelectric Complex, aiming to improve dam monitoring and anomaly detection. The approach offers advantages such as enhanced vegetation analysis, reduced need for manual inspections, and increased efficiency in identifying potential structural risks. However, it would be beneficial to revise the manuscript based on the following comments to improve its clarity, robustness, and applicability.
1. How does the proposed Random Forest (RF) model compare to deep learning-based segmentation models like U-Net or DeepLab in terms of accuracy and generalization?
2. What are the specific limitations of using multispectral imagery from UAVs in different environmental conditions, such as cloud cover, seasonal vegetation changes, or varying light conditions?
3. How would the model perform if applied to different types of embankment dams outside the Belo Monte Hydroelectric Complex?
4. Did you consider potential overfitting in their feature engineering process, especially given the high classification accuracy?
5. What are the implications of omitting the Panchromatic band from the analysis, and how might its inclusion affect classification performance?
6. How does the resolution of the UAV imagery impact the ability of the model to detect fine-scale features like small cracks or minor vegetation anomalies?
7. What is the computational cost of deploying this methodology on a large-scale dam monitoring system, and how feasible is real-time processing?
8. How reliable are the manually annotated segmentation masks, and did the authors conduct an inter-annotator agreement analysis to assess annotation consistency?
9. Did the authors consider potential bias in their dataset, given that the training data is based on only two dam structures?
10. How do the findings translate into actionable recommendations for dam safety monitoring teams, and what are the next steps for implementation in real-world scenarios?
11. The paper does not compare the RF model’s performance against deep learning-based approaches, which are widely used in semantic segmentation.
12. There is no discussion on how well the model generalizes to other embankment dams with different geological or environmental characteristics.
13. The authors do not elaborate on the potential challenges of UAV-based data collection, such as battery life, flight stability, or regulatory restrictions.
14. The effect of varying weather conditions (e.g., cloud cover, humidity, rainfall) on image quality and segmentation performance is not addressed.
15. The study does not explore how removing or adding specific features impacts model performance.
16. The study focuses on static images but does not address how the model could be applied to track changes over time.
17. There is no discussion on how this approach could be integrated into an operational dam monitoring workflow.
18. The training dataset is derived from only two dam structures, which may not represent broader dam conditions.
19. The authors do not provide an in-depth discussion of the types of errors the model makes and potential reasons for them.
20. The study does not explore the feasibility of scaling this approach to monitor multiple large dams in real-time.
21. To further strengthen the manuscript, the authors may consider incorporating additional relevant studies that align with their research.
These studies provide useful perspectives on remote sensing, machine learning, and dam stability analysis, which could further support the discussion and contextualization of the findings. Integrating relevant references may enhance the depth of the manuscript and its contribution to the field.

---

## Round 0.2 · accepted · Accept

The authors have revised the manuscript and addressed the reviewers’ concerns.

Reviewer 3 ·

Basic reporting

accept

Experimental design

accept

Validity of the findings

accept